# Saliva SARS-CoV-2 Antibody Prevalence in Children

Maya W. Keuning,[a] Marloes Grobben,[b] Anne-Elise C. de Groen,[a] Eveline P. Berman-de Jong,[c] Merijn W. Bijlsma,[c] Sophie Cohen,[c] Mariet Felderhof,[d] Femke de Groof,[e] Daniel Molanus,[f] Nadia Oeij,[f] Maarten Rijpert,[g] Hetty W. M. van Eijk,[b] Gerrit Koen,[b] Karlijn van der Straten,[b] Melissa Oomen,[b] Remco Visser,[h,i] Federica Linty,[h,i] Maurice Steenhuis,[i,j] Gestur Vidarsson,[h,i] Theo Rispens,[i,j] Frans B. Plötz,[c,k] Marit J. van Gils,[b] Dasja Pajkrt[a]

[a]Department of Pediatric Infectious Diseases, Rheumatology, & Immunology, Emma Children's Hospital, Amsterdam UMC, University of Amsterdam, Amsterdam, The Netherlands

[b]Department of Medical Microbiology and Infection Prevention, Amsterdam Institute of Infection and Immunity, Amsterdam UMC, University of Amsterdam, Amsterdam, The Netherlands

[c]Department of Pediatrics, Emma Children's Hospital, Amsterdam UMC, University of Amsterdam, Amsterdam, The Netherlands

[d]Department of Pediatrics, Flevoziekenhuis, Almere, The Netherlands

[e]Department of Pediatrics, Noordwest Ziekenhuisgroep, Alkmaar, The Netherlands

[f]Department of Pediatrics, Amstellandziekenhuis, Amstelveen, The Netherlands

[g]Department of Pediatrics, Zaans Medisch Centrum, Zaandam, The Netherlands

[h]Department of Experimental Immunohematology, Sanquin Research, Amsterdam, The Netherlands

[i]Landsteiner Laboratory, Amsterdam UMC, University of Amsterdam, Amsterdam, The Netherlands

[j]Department of Immunopathology, Sanquin Research, Amsterdam, The Netherlands

[k]Department of Pediatrics, Tergooi Hospital, Blaricum, The Netherlands

Maya W. Keuning and Marloes Grobben contributed equally to this article. Author order was determined on the basis of contributions to study conception and design.

**ABSTRACT**  COVID-19 patients produce circulating and mucosal antibodies. In adults, specific saliva antibodies have been detected. Nonetheless, seroprevalence is routinely investigated, while little attention has been paid to mucosal antibodies. We therefore assessed SARS-CoV-2-specific antibody prevalence in serum and saliva in children in the Netherlands. We assessed SARS-CoV-2 antibody prevalence in serum and saliva of 517 children attending medical services in the Netherlands (irrespective of COVID-19 exposure) from April to October 2020. The prevalence of SARS-CoV-2 spike (S), receptor binding domain (RBD), and nucleocapsid (N)-specific IgG and IgA were evaluated with an exploratory Luminex assay in serum and saliva and with the Wantai SARS-CoV-2 RBD total antibody enzyme-linked immunosorbent assay in serum. Using the Wantai assay, the RBD-specific antibody prevalence in serum was 3.3% (95% confidence interval [CI]. 1.9 to 5.3%). With the Luminex assay, we detected heterogeneity between antibodies for S, RBD, and N antigens, as IgG and IgA prevalence ranged between 3.6 and 4.6% in serum and between 0 and 4.4% in saliva. The Luminex assay also revealed differences between serum and saliva, with SARS-CoV-2-specific IgG present in saliva but not in serum for 1.5 to 2.7% of all children. Using multiple antigen assays, the IgG prevalence for at least two out of three antigens (S, RBD, or N) in serum or saliva can be calculated as 3.8% (95% CI, 2.3 to 5.6%). Our study displays the heterogeneity of the SARS-CoV-2 antibody response in children and emphasizes the additional value of saliva antibody detection and the combined use of different antigens.

**IMPORTANCE**  Comprehending humoral immunity to SARS-CoV-2, including in children, is crucial for future public health and vaccine strategies. Others have suggested that mucosal antibody measurement could be an important and more convenient tool to evaluate humoral immunity compared to circulating antibodies. Nonetheless, seroprevalence is routinely investigated, while little attention has been paid to mucosal antibodies. We show the heterogeneity of SARS-CoV-2 antibodies, in terms of

Address correspondence to Maya W. Keuning, m.w.keuning@amsterdamumc.nl.

Comprehending humoral immunity to SARS-CoV-2 is crucial for future public health and vaccine strategies. Our study displays the heterogeneity of the SARS-CoV-2 antibody response in children and emphasizes the value of saliva antibody detection.

both antigen specificity and differences between circulating and mucosal antibodies, emphasizing the additional value of saliva antibody detection next to detection of antibodies in serum.

**KEYWORDS** SARS-CoV-2, antibodies, children, humoral immunity, prevalence, saliva

Severe acute respiratory syndrome coronavirus 2 (SARS-CoV-2) is a positive-sense single-stranded RNA virus from the Coronaviridae family causing coronavirus disease 2019 (COVID-19). In 2020, this novel virus emerged as the cause of a pandemic. In the Netherlands, the first COVID-19 patient was confirmed on 27 February 2020. After the first peak in hospital admission rates during March and April 2020, the Netherlands endured a second peak during the second half of 2020. Epidemiological data on immunity are essential to understand disease pathology and to guide national prevention measures and possibilities for vaccine development (1).

Generally, humoral immunity is measured as the presence of pathogen-specific antibodies in serum. The prevalence of SARS-CoV-2 antibodies in serum has been described in several countries, including the Netherlands (2–4), with a potential durability of over 8 months (5, 6). Although the mucosa of the upper respiratory tract is the primary entry site of SARS-CoV-2, little attention has been paid to the presence of mucosal antibodies as part of the humoral immune response (7). For other viruses such as hepatitis B virus, norovirus, and human immunodeficiency virus type 1, studies have shown high similarity in circulating and mucosal antibody profiles, advocating for the use of saliva samples to measure humoral immunity (8–10). In respiratory syncytial virus (RSV), mucosal anti-RSV IgA and IgG combined proved at least as reliable as serum to detect infection. These saliva antibodies could help to distinguish current RSV (re-)infection from a false-positive result due to preexisting maternal antibodies (11). In adult COVID-19 patients, promising results on SARS-CoV-2-specific saliva antibodies with neutralizing capacities and a durability similar to serum have been reported (12–14). Interestingly, in asymptomatic or mild COVID-19, mucosal antibodies were detected, even in some seronegative patients (15). Mucosal antibody measurement could be an important and more convenient tool to evaluate humoral immunity in children, since they often have asymptomatic or mild disease (16, 17).

Among children, circulating antibody prevalence ranges from 0.9% in the United States to 7.3% in Switzerland (16, 18). The use of saliva antibody assays has yet to be explored in asymptomatic cases and in a pediatric population. Thus, the primary aim of this study was to evaluate the SARS-CoV-2-specific antibody prevalence in saliva compared to serum in a pediatric population during the COVID-19 outbreak in 2020 in the Netherlands. Since the Luminex assay provides a very sensitive method which is easily adjustable to different antigens, antibody isotypes, and sample types, we utilized this explorative assay in addition to the validated Wantai assay.

## RESULTS

**Study population.** A total of 589 children were approached, 517 of which were included (see Fig. S1 in the supplemental material). The characteristics of the participants are shown in Table S2. Figure 1A shows the age distribution across the inclusion period. The median age was 11 years (interquartile range [IQR], 5 to 15 years). An immunocompromised state and an underlying illness were described in 35.8% and 24.8% of children, respectively, while 38.9% did not have a relevant medical history. Sex was equally distributed among comorbidity groups (Fig. 1B). SARS-CoV-2 PCR on nasopharyngeal swabs had been performed previously by the treating physician in 107 children, either due to clinical suspicion of COVID-19 or as preprocedural testing. Paired serum and saliva samples of sufficient volume for all assays were available for 413/517 (82%) children, with a median age of 12 years (IQR, 7 to 15 years).

**Antibody prevalence.** Figure 2 shows the estimated prevalence determined by the Wantai receptor binding domain (RBD) total antibody assay (Fig. 2A) and the Luminex assay for spike (S), RBD, and nucleocapsid (N)-specific IgG (Fig. 2B) and IgA (Fig. 2C) in

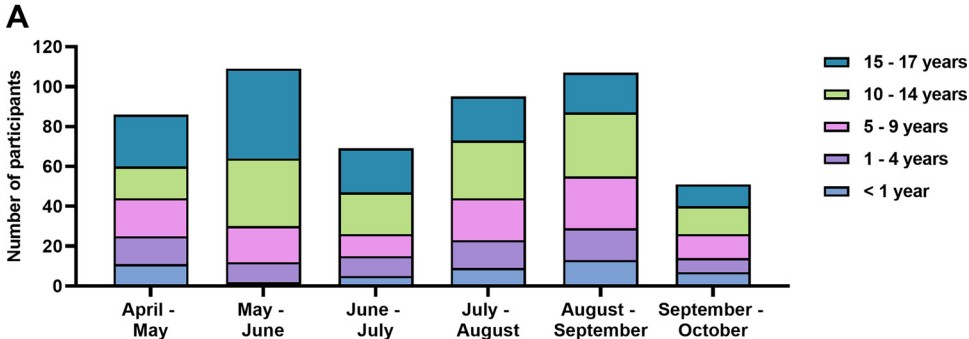

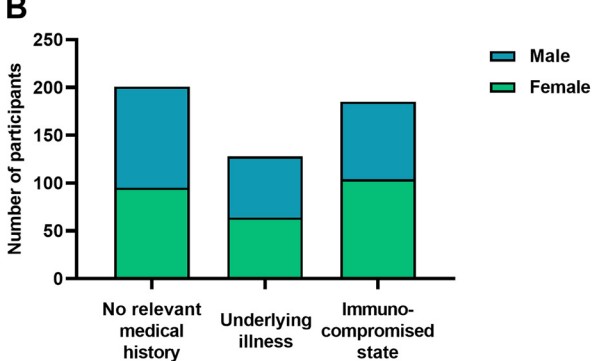

**FIG 1** Age distribution, comorbidity, and sex. (A) The distribution of age across the inclusion period of 24 weeks is depicted for the total number of cases ($n$ = 517), with each bar representing 4 weeks. (B) Comorbidity and male/female ratio of the study sample is calculated for the nonmissing values ($n$ = 514, excluding 3 missing values for comorbidity).

serum and saliva. Titers from the Luminex assay are shown in Fig. S2, and the assay cut-off and assay performance parameters are in Table S1. The prevalence of RBD-specific antibodies in serum was 16/487 (3.3%; 95% confidence interval [CI], 1.9 to 5.3%) with the Wantai assay. With the Luminex assay, the prevalence of SARS-CoV-2-specific antibodies in serum ranged between 3.3% (95% CI, 1.9 to 5.2%) and 4.3% (95% CI, 2.7 to 6.4%) depending on the antigen and isotype measured. The prevalence of SARS-CoV-2-specific antibodies in saliva ranged between 0.0% (95% CI, 0.0 to 0.7%) and 4.4% (95% CI, 2.7 to 6.8%).

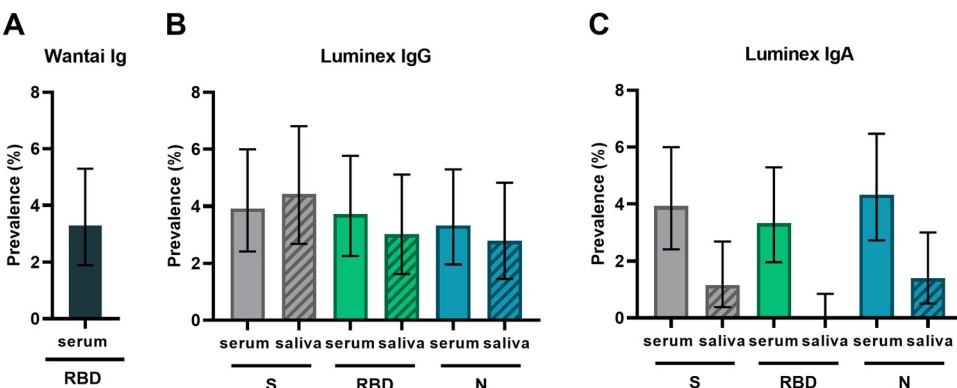

**FIG 2** SARS-CoV-2 antibody prevalence estimates in serum and saliva. (A) Prevalence estimate of SARS-CoV-2 RBD total antibodies in the Wantai assay for serum. (B and C) Prevalence estimates of SARS-CoV-2 S- (gray bars), RBD- (green bars), and N-specific (blue bars) IgG (B) and IgA (C) in the Luminex assays for serum (solid bars) and saliva (hatched bars). Prevalence was the calculated proportion with a value above the determined cutoff out of nonmissing values. Estimates are shown with 95% confidence intervals. S, trimeric SARS-CoV-2 spike protein; RBD, the monomeric receptor binding domain of the SARS-CoV-2 spike protein; N, SARS-CoV-2 nucleocapsid protein.

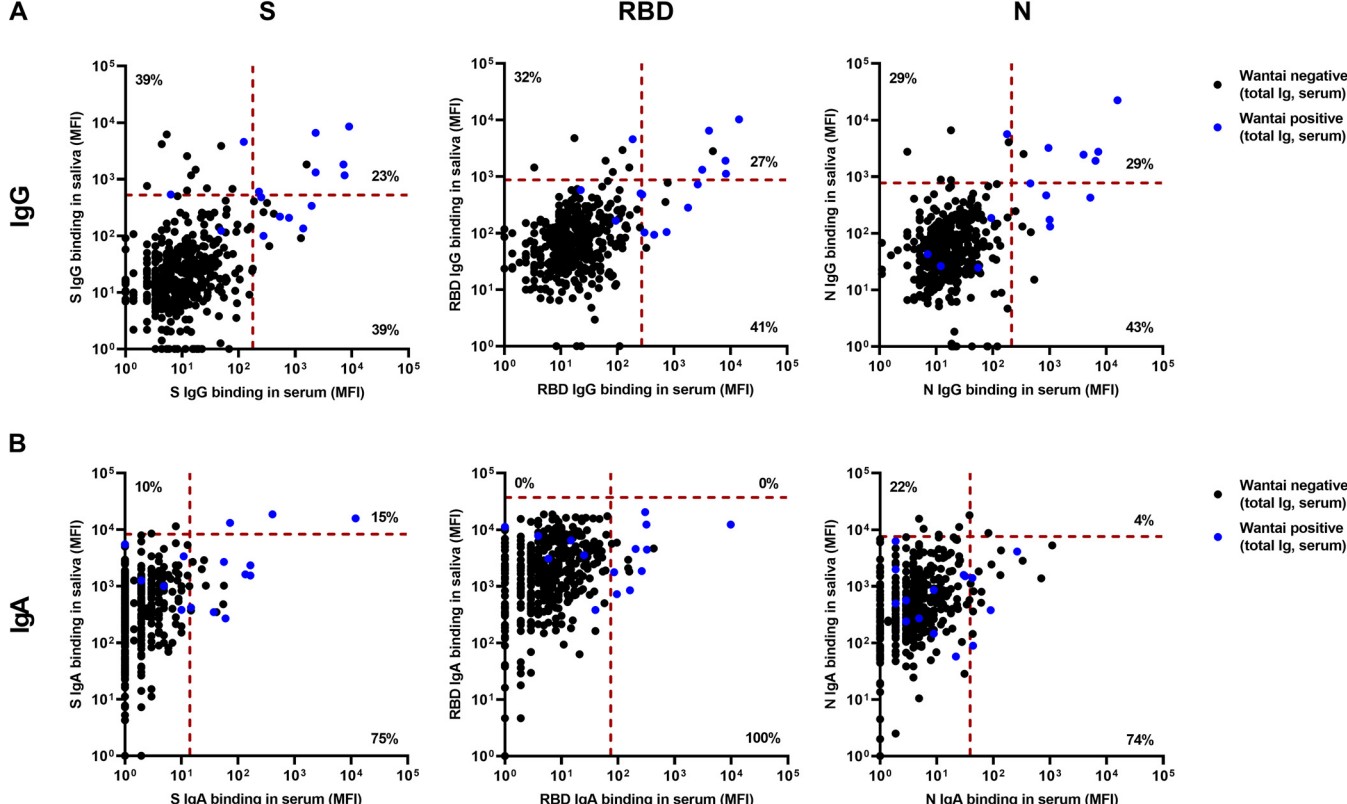

**FIG 3** Correspondence of Luminex assays for serum and saliva. (A and B) SARS-CoV-2 S-, RBD-, and N-specific IgG (A) and IgA (B), measured in paired serum and saliva samples (n = 413) by Luminex assay, expressed as MFI. Only samples also measured in the Wantai assay are shown, with the positive Wantai results indicated in blue. Serum and saliva are plotted against each other to reveal the differences between each compartment. The red dotted lines are the cutoff values to discriminate positive and negative measurements in the Luminex assays. Percentages represent data points which are positive for both compartments or for a single compartment as the percentage of total positives in each graph. S, trimeric SARS-CoV-2 spike protein; RBD, protein of only the monomeric receptor binding domain of the SARS-CoV-2 spike protein; N, SARS-CoV-2 nucleocapsid protein; MFI, median fluorescence intensity.

The prevalence of SARS-CoV-2-specific antibodies did not increase over calendar months in the Luminex assays or the Wantai assay (Fig. S3). The seroprevalence was 4/200 (2.0%) in children aged 0 to 10 years and 12/287 (4.2%) in children aged 10 to 17 years in the Wantai assay (Table S2). The seroprevalence was 3.3% in children with an immunocompromised state, 5.0% in children with underlying illness, and 2.2% in children with no relevant medical history. Most of the children that were positive in the Wantai assay (13/16, 81%) had COVID-19 symptoms at the time of inclusion or in the previous 4 weeks or reported a COVID-19-positive household member.

**Comparison of serum and saliva SARS-CoV-2 antibody prevalence.** In the Luminex assay, 31/422 (7.4%) children had detectable S-specific IgG, while 22/422 (5.2%) had detectable RBD-specific IgG, and 21/422 (5.0%) had detectable N-specific IgG in serum and/or saliva (Fig. 2B). Figure 3A shows the correspondence between serum and saliva IgG for all paired samples in the Luminex and Wantai assays. Between 7/31 and 6/21 (23 to 29%) of these children showed corresponding positive titers in both serum and saliva, depending on the antigen used. However, 12/31 to 9/21 (38 to 43%) children had positive SARS-CoV-2-specifc IgG titers in serum but not in saliva, and 6/21 to 12/31 (29 to 39%) had positive titers in saliva but not in serum.

Comparable with these findings, between 6/15 and 8/15 (40 to 53%) Wantai positive children also had positive saliva IgG titers in the Luminex assay, but between 6/12 and 11/19 (50 to 58%) children with SARS-CoV-2-specific saliva IgG in the Luminex assay were negative in the Wantai assay. This corresponds to 6 to 11/413 (1.5 to 2.7%) of the total paired sample, depending on the antigen used.

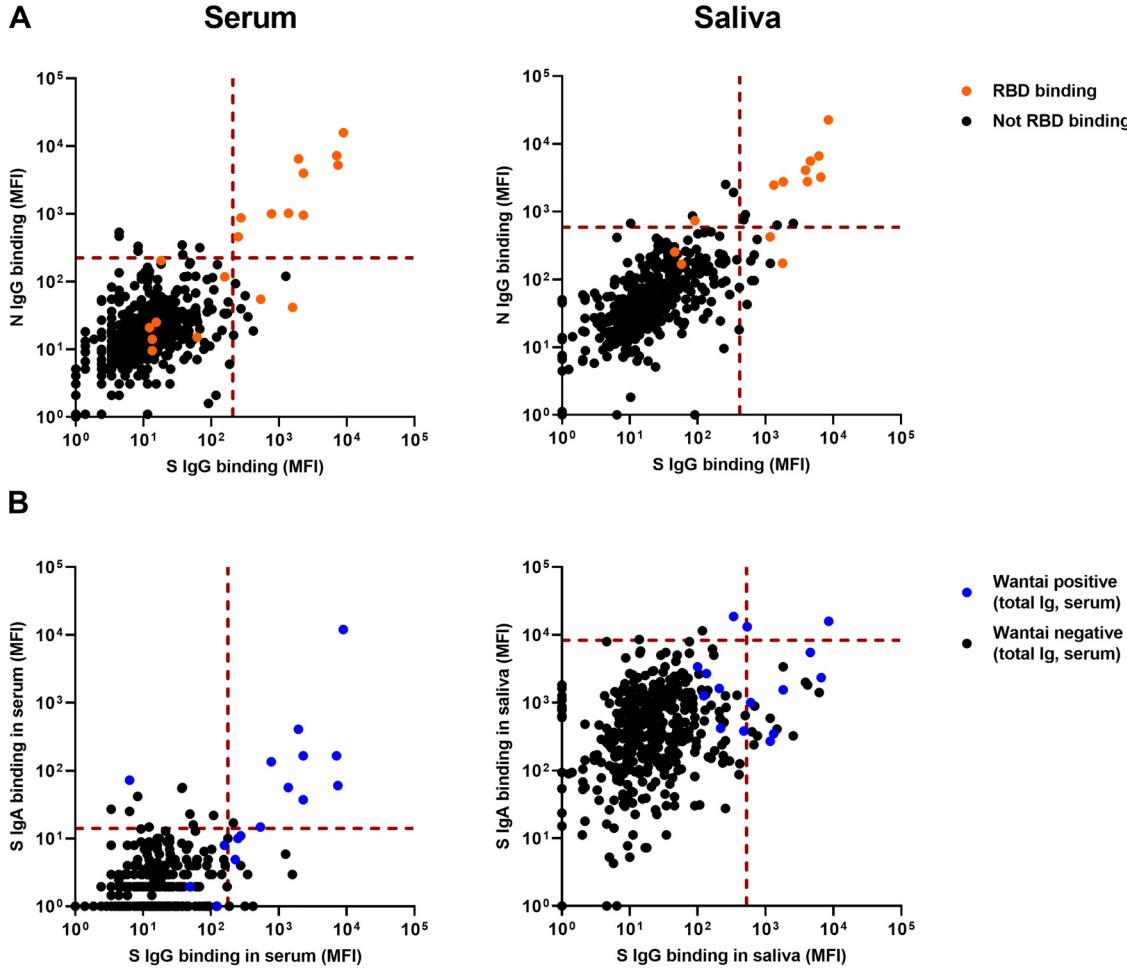

**FIG 4** Correspondence of Luminex assays for different antigens and isotypes. (A) S-, RBD-, and N protein-specific IgG, measured in serum (*n* = 509, left) and saliva (*n* = 430, right) in the Luminex assay, expressed as MFI. S and N are plotted against each other, and RBD-positive samples are shown in orange. The red dotted lines are the cutoff values to discriminate positive and negative measurements for S and N. (B) SARS-CoV-2 S-specific IgG and IgA in serum (*n* = 487, left) and saliva (*n* = 413, right) measured in the Luminex assay, expressed as MFI. IgG and IgA are plotted against each other to reveal the correspondence between the two isotypes. Samples that were also positive in the Wantai RBD total antibody assay are shown in blue. S, trimeric SARS-CoV-2 spike protein; RBD, monomeric receptor binding domain of the SARS-CoV-2 spike; N, SARS-CoV-2 nucleocapsid protein; MFI, median fluorescence intensity.

We compared IgA responses in serum and saliva (Fig. 3B), but there was low agreement between the two compartments (0/15 to 3/20; 0 to 15% with detectable IgA in both serum and saliva). The Luminex IgA assays were positive in saliva while negative in serum in 0/15 to 5/23 (0 to 22%) of children with IgA antibodies.

**Combined Luminex assay.** Figure 4A shows the correspondence between the three SARS-CoV-2-specific antigens. All children with S- and N-specific IgG in serum also had positive titers of RBD-specific IgG. Comparing IgG and IgA antibodies (Fig. 4B) in serum, a few children showed S-specific IgA but not IgG antibodies (10/509; 2.0%). All but one child positive for both serum IgG and IgA in the Luminex assay were also positive in the Wantai total antibody assay (9/487; 1.8%). There was low correspondence between IgG and IgA in saliva.

We explored combining multiantigen assays to calculate the SARS-CoV-2 IgG prevalence in several ways (Fig. 5). The total prevalence of IgG binding to any of the antigens is 34/509 (6.7%) in serum and 26/430 (6.0%) in saliva. To increase the specificity, the IgG prevalence for at least two out of three antigens (S, RBD, or N protein) can be calculated as 2.4% for serum, 2.3% for saliva, and 3.8% (95% CI, 2.3 to 5.6%) when both

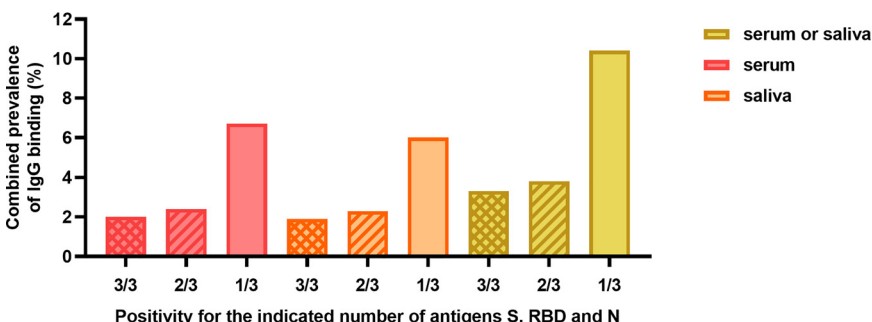

**FIG 5** Combined SARS-CoV-2 IgG antibody prevalence. Combined prevalence of the explorative Luminex assay of IgG in serum (pink bars), in saliva (orange bars), or in serum and saliva combined (yellow bars). The combined prevalence was calculated for children positive for 3/3 SARS-CoV-2 antigens (crosshatched bars) at least 2/3 antigens (hatched bars), and for children positive for at least 1/3 antigens (solid bars). S, trimeric SARS-CoV-2 spike protein; RBD, monomeric receptor binding domain of the SARS-CoV-2 spike; N, SARS-CoV-2 nucleocapsid protein.

serum and saliva are measured. Considering only children with positive Luminex titers for two out of three antigens in saliva, 4/413 children had saliva antibodies, while they were negative in the Wantai assay, corresponding to 1.0% (95% CI, 0.3 to 2.2%) of the total sample.

## DISCUSSION

This is the first study evaluating concurrent mucosal and circulating SARS-CoV-2-specific antibodies in a large pediatric cohort. We found children with detectable SARS-CoV-2-specific saliva IgG, while circulating antibody levels (measured with the Wantai assay) were undetectable. Similarly, not all seropositive children had detectable saliva IgG. Additionally, we detected heterogeneity in the presence of antibodies to different SARS-CoV-2-specific antigens.

We found an antibody seroprevalence comparable to that of the S-specific IgG seroprevalence among 1,000 Dutch children during June and July 2020 in a population-based study (19, 20). Most children in our population had an underlying illness or immunocompromised state. Therefore, the similarities to national seroprevalence studies are particularly interesting and suggest similar COVID-19 incidence in our hospital-based cohort compared to the general community. Similar to the Netherlands, other European countries report lower seroprevalence in children (range, 0.8 to 7.3%) compared to adults (range, 1 to 20%) (18, 21).

Although many reports are appearing on SARS-CoV-2 seroprevalence, the initial confrontation of SARS-CoV-2 with the adaptive immune system is located at the mucosal surface of the respiratory tract. To our knowledge, only a few adult cohorts report on the associations between circulating and mucosal SARS-CoV-2-specific antibodies. For example, a sensitivity of 84.2% and a specificity of 100% for saliva S-specific IgG to detect PCR- and seropositive patients in a symptomatic population was reported (12). The durability appears similar to that of circulating antibodies, as salivary antibodies were shown to be measurable for up to 9 months after infection in convalescent mild COVID-19 patients (22). The findings from three cohorts of SARS-CoV-2-infected adults suggest that mucosal IgG antibodies can be used as a surrogate for circulating IgG due to the high similarity (12–14).

However, a proportion of seronegative children in our sample did have saliva antibodies (mostly IgG). Corroborating these findings, 15 to 20% of seronegative health care workers had mucosal SARS-CoV-2 S-specific antibodies with, in some cases, comparable *in vitro* neutralizing capacity to serum (15, 23). A possible explanation for the discordance between circulating and mucosal antibodies could be the disease severity. Cervia et al. hypothesize that the mucosal antibody response is more prominent in

mild infection and in younger individuals, although their youngest participant was 30 years old (15). Similarly, a preprint validation study of a saliva IgA assay observed asymptomatic individuals with saliva SARS-CoV-2-specific antibodies despite a negative PCR and/or serum antibody tests (24). Hence, reporting seroprevalence alone may result in an underestimation of humoral immunity, particularly in younger and mildly infected patients.

We still lack full understanding of how IgA provides additional value for evaluating humoral immunity (25). IgA can appear earlier in blood than IgG following SARS-CoV-2 infection (26). Although IgA is the key immunoglobulin for mucosal immunity, evidence of SARS-CoV-2-specific saliva IgA is inconsistent. Compared to IgG, saliva IgA is less correlated with serum IgA (3). Saliva IgG is mostly derived from circulatory IgG through transudation, whereas saliva IgA can be produced locally (27). This is also observed in vaccination response studies, showing saliva S-specific IgG in all fully vaccinated participants, while only 60% showed saliva S-specific IgA (28). Consistent with our findings, MacMullan et al. and Pisanic et al. emphasize the lower sensitivity for IgA to detect PCR-positive patients compared to IgG (12, 13). The high variation in mucosal IgA titers complicates detection of SARS-CoV-2-specific saliva IgA, possibly caused by polyreactive IgA known to function as a nonspecific mucosal immune barrier (24).

There is growing evidence on the use of SARS-CoV-2 multiantigen assays in epidemiological surveys. In line with other studies, we observed that positive individuals often do not show equally elevated titers across all three SARS-CoV-2-specific antigens. In an epidemiological survey of 1,225 blood donors, seroprevalence with single-antigen assays also varied widely between 0.8% and 7.5% depending on the antigen type (29). A potential association between specific antigens and disease severity has been proposed. Outpatients with mild or asymptomatic infection showed higher ratios of S- and RBD-specific antibodies than of N-specific antibodies, whereas all three antigens were effective for detecting responses in hospitalized patients (23, 30). Considering their different kinetics (31) and functions in the humoral immune response and the broad clinical spectrum of COVID-19, combining multiple SARS-CoV-2 antigens seems an appropriate method when investigating population prevalence (32).

Targeting multiple antigens in multiple compartments to evaluate the humoral response can increase sensitivity but may consequently increase false positives. A golden standard is unfortunately still missing to determine true rates, as both PCR and serology can have false-negative results (33, 34). To increase the certainty of SARS-CoV-2 exposure and minimize the possibility of a false-positive result (particularly in a presumed low-prevalence context) (35), we could define a sample as positive only when the IgG level is above the cutoff for at least two antigen assays as proposed in other studies (13, 31, 32, 36). In adult cohorts of (symptomatic) PCR-proven COVID-19 patients and pre-COVID-19 era controls, combining multiple N and S/RBD antibody assays resulted in higher diagnostic accuracy (13, 32, 36, 37). Although Luminex assays in serum are known to have high accuracy to detect previous SARS-CoV-2 infection (31, 32, 36, 37) and the validated Wantai assay provides the context for the results, follow-up research should focus on validation of established assays on materials other than serum as well as the combined use of different antigens in high- and low-prevalence settings.

Our study encountered several other limitations. We did not evaluate the potential moment of infection; thus, we could have missed SARS-CoV-2-exposed children, as IgG can be detected after 10 days and for several months post-symptom onset, while IgM and IgA wane more quickly (3). We minimized this effect by detecting multiisotype antibodies. Moreover, cross-reactive antibodies from previous coronavirus types could have resulted in positivity without a history of actual SARS-CoV-2 infection. Antibody cross-reactivity is a known phenomenon for pathogens with shared structural motifs (38, 39). In a cohort sampled in the pre-COVID-19 era, detectable SARS-CoV-2 S-specific antibodies were found and at a higher frequency in children than adults, peaking to 62% between 6 and 16 years of age (39). Although these antibody titers were lower

than those in COVID-19 patients, their sera exhibited neutralizing activity against SARS-CoV-2 (39). Thus, even if some of the antibody titers we found are the result of cross-reactivity, they could still be functional. Since we did not perform *in vitro* neutralization testing, the functionality of the (mucosal) antibodies in our cohort against SARS-CoV-2 remains unknown.

**Conclusion and implications.** Comprehending humoral immunity to SARS-CoV-2, including in children, is crucial for future public health and vaccine strategies. We therefore detected SARS-CoV-2 antibody prevalence in serum and saliva of children. Our study displays the heterogeneity of the SARS-CoV-2 antibody response in children and emphasizes the additional value of saliva assays for antibody detection as well as the combined use of different antigens. Validation of multiantigen saliva antibody assays is recommended for further research.

## MATERIALS AND METHODS

**Study design and participants.** This prospective cross-sectional study included simultaneous convenience blood and saliva sampling of children attending medical services at one of seven participating secondary- and tertiary-care hospitals in the Northwest region of the Netherlands during 24 consecutive weeks (12 April to 2 October 2020). Inclusion criteria were children aged 0 to 18 years residing in the Netherlands who required blood testing or intravenous cannulation for any reason. Eligibility was irrespective of (suspected) acute or prior COVID-19 infection. Children were excluded if sample collection of neither serum nor saliva was sufficient.

We recorded age, sex, COVID-19-related symptoms, and proven or suspected COVID-19 in household members. COVID-19-related symptoms were defined as fever (>37.5°C), sore throat, cough, shortness of breath, tachypnea, headache, abdominal cramps, and diarrhea. Electronic patient files were used to extract previous SARS-CoV-2 PCR assay results and medical history that could influence the humoral immune response or infection severity. Children with immunodeficiency, autoimmune disease, hematological malignancies, and/or use of immunomodulating drugs were defined as having an "immunocompromised state." Immunomodulating drugs included azathioprine, methotrexate, monoclonal antibodies, immunoglobulins, and corticosteroids. We defined children with an "underlying illness" as children with obesity (body mass index [BMI], ≥30); respiratory, cardiovascular, endocrine, metabolic, hematologic, or kidney diseases; solid malignancies; or psychomotor retardation. Previously healthy children were categorized as having "no relevant medical history." The study protocol was approved by the ethics committee of the Amsterdam University Medical Centers (NL73556.018.20) and conducted in accordance with good clinical practice standards. Written informed consent was obtained from both parents/guardians and/or from children above the legal age of consent.

**Serum and saliva sampling.** Saliva was obtained using a sterile container or a buccal swab (ORACOL saliva collection device, product code S10; Malvern Medical Developments Ltd.). Saliva samples were either stored at −70°C until centrifuging or directly centrifuged (at 1,000 rpm for 10 min). Supernatant and pellets were stored in aliquots at −80°C. During venipuncture, a blood sample of 1 to 5 ml was collected and centrifuged, and was serum stored at −20°C.

**Serum assays.** Seroprevalence was assessed with the FDA-approved Wantai SARS-CoV-2 RBD total antibody enzyme-linked immunosorbent assay following the manufacturer's instructions, with a sensitivity of 96.7% (95% CI, 83.3 to 99.4%) and specificity of 97.5% (95% CI, 91.3 to 99.3%) (40), and confirmed with an in-house-developed SARS-CoV-2 RBD total antibody bridge assay as described previously (sensitivity, 98.1%; specificity, 99.5%) (41).

**Protein coupling to Luminex beads.** An explorative Luminex assay was developed to investigate antigen-specific IgG and IgA in serum and saliva. A recombinant prefusion ectodomain trimer of SARS-CoV-2 S protein, the monomeric RBD of the S protein, and N protein were designed, produced, and purified as previously described (42, 43). The proteins were covalently coupled to Magplex beads (Luminex) using a two-step carbodiimide reaction at a ratio of 75 $\mu$g protein to 12.5 million beads for S, at an equimolar concentration for N, and at 3× the equimolar concentration for RBD. Beads were washed with 100 mM monobasic sodium phosphate, pH 6.2, activated with sulfo-*N*-hydroxysulfosuccinimide and 1-ethyl-3-(3-dimethylaminopropyl) carbodiimide (Thermo Fisher Scientific), and incubated for 30 min on a rotator at room temperature (RT). Activated beads were washed 3 times with 50 mM MES (morpholineethanesulfonic acid), pH 5.0, proteins were added, and beads were incubated for 3 h on a rotator at RT. The beads were washed with phosphate-buffered saline (PBS) and blocked with PBS containing 2% bovine serum albumin (BSA), 3% fetal calf serum, and 0.02% Tween 20 at pH 7.0 (PBS-blocking) for 30 min on a rotator at RT. Finally, the beads were washed, stored in PBS containing 0.05% sodium azide at 4°C, and used within 6 weeks.

**Luminex assays.** A total of 50 $\mu$l of PBS-blocking containing 20 of each bead per $\mu$l was incubated with 50 $\mu$l of 1:10,000 diluted serum or 1:10 diluted saliva supernatant in PBS-blocking overnight on a plate shaker at 4°C. Subsequently, plates were washed with Tris-buffered saline (TBS) containing 0.05% Tween 20 (TBST) using a magnetic separator. Beads were resuspended in 50 $\mu$l of goat anti-human IgG-PE (Southern Biotech) or goat anti-human IgA-PE (Southern Biotech) in PBS-blocking and incubated on a plate shaker at RT for 2 h. Afterward, beads were washed with TBST and resuspended in 70 $\mu$l MAGPIX drive fluid (Luminex). Read-out was performed on a MAGPIX system (Luminex). The resulting median

fluorescence intensity (MFI) values per bead were background-corrected by subtraction of the MFI values of buffer only. A titration of serum and saliva of an adult convalescent COVID-19 patient was used to normalize data between plates. The cutoff was determined at 2 (geometric) standard deviations above the geometric mean of the total sample ($n = 509$ for serum, $n = 430$ for saliva), separately for each combination of antigen, sample type and secondary antibody (Table S1). Beads with no antigen were included to confirm the absence of antibodies binding to beads or blocking components for each individual sample. Prepandemic serum pools ($n = 3$) or healthy donor saliva (early in the pandemic with negative PCR and/or absence of symptoms; $n = 4$) were included on each assay plate as negative controls. The replicability was calculated on samples measured twice, and the assay precision was calculated using positive-control sera ($n = 6$) or saliva ($n = 5$) included on each plate (Table S1).

**Statistical analyses.** We estimated the prevalence of circulating and mucosal antibodies as the proportion of children with a result above the cutoff value. Confidence intervals were calculated with the Clopper-Pearson method in IBM SPSS Statistics version 26 predictive analytics software. In accordance with the World Health Organization's Investigation Protocol for COVID-19 seroepidemiological research, we stratified participants into predefined age groups as follows: <1 year, 1 to 4 years, 5 to 9 years, 10 to 14 years, and 15 to 17 years (1). We calculated a minimum sample size of 139 children per age group (0 to 1 year, 1 to 5 years, and 5 to 18 years) to detect a seroprevalence of 10% (95% CI, 5 to 15%).

## SUPPLEMENTAL MATERIAL

Supplemental material is available online only.

**SUPPLEMENTAL FILE 1**, PDF file, 0.6 MB.

## ACKNOWLEDGMENTS

This study would not have been possible without the instrumental and passionate support of Stichting Steun Emma Kinderziekenhuis, of the physicians, laboratory, and nursing staff of all participating care centers, the AUMC microbiology staff, the student team for inclusion of patients and the student team for data entry, the donors of control samples, all participants and their caregivers, and the creative contribution of Eli Vlessing.

This work was supported by the Contribute Foundation.

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
