## [Reviewer comments · Microbiology Spectrum]

Microbiology Spectrum

Saliva SARS-CoV-2 antibody prevalence in children

Maya Keuning, Marloes Grobben, Anne-Elise de Groen, Eveline Berman-de Jong, Merijn Bijlsma, Sophie Cohen, Mariet Felderhof, Femke de Groof, Daniel Molanus, Nadia Oeij, Maarten Rijpert, Hetty van Eijk, Gerrit Koen, Karlijn van der Straten, Melissa Oomen, Remco Visser, Federica Linty, Maurice Steenhuis, Gestur Vidarsson, Theo Rispens, Frans Plötz, Marit van Gils, and Dasja Pajkrt

Corresponding Author(s): Maya Keuning, Amsterdam UMC

Review Timeline:

Submission Date:	June 30, 2021
Editorial Decision:	August 2, 2021
Revision Received:	August 11, 2021
Accepted:	August 12, 2021

Editor: Jennifer Dien Bard

Reviewer(s): The reviewers have opted to remain anonymous.

Transaction Report:

DOI: <https://doi.org/10.1128/Spectrum.00731-21>

August 2, 2021

Dr. Maya W. Keuning
Amsterdam UMC
Amsterdam
Netherlands

Re: Spectrum00731-21 (Saliva SARS-CoV-2 antibody prevalence in children)

Dear Dr. Maya W. Keuning:

Thank you for submitting your manuscript to Microbiology Spectrum. Your manuscript has now been reviewed by two expert reviewers. Based on these comments (attached below), as well as my own review, I have decided that your manuscript is potentially suitable for publication, subject to revisions being made.

When submitting the revised version of your paper, please provide (1) point-by-point responses to the issues raised by the reviewers as file type "Response to Reviewers," not in your cover letter, and (2) a PDF file that indicates the changes from the original submission (by highlighting or underlining the changes) as file type "Marked Up Manuscript - For Review Only". Please use this link to submit your revised manuscript - we strongly recommend that you submit your paper within the next 60 days or reach out to me. Detailed information on submitting your revised paper are below.

Link Not Available

Sincerely,

Jennifer Dien Bard

Journals Department
Reviewer comments:

Reviewer #1 (Comments for the Author):

Thank you for this paper it was very interesting. When you compare your seroprevalence rates with the national seroprevalence study it may help to expand on the fact that their study was community based rather than your hospital based study. Your sample is very different as you have recruited from a hospital population rather than a representative paediatric population. The fact that you are seeing similar results both in the community and the hospital based seroprevalence studies is interesting. I think you make a good case for saliva antigens to be used in future research especially since this is more likely to be acceptable to parents of younger children who may be put off by a blood test.

Reviewer #2 (Comments for the Author):

Dear authors,

Thank you very much for this interesting article.

The usefulness of saliva for diagnosing SARS-CoV-2 seroprevalence in children is a key for this population age-group, and the discordances with serum are a matter of concern.

However, I would like to share with you some doubts about the manuscript.

Abstract: it is difficult to read, the information in the results need to be improved, i.e. lines 51 and 52 cannot begin with numbers of percentages.

Introduction: why do you comment the incidence of COVID-19 in December if the study was carried out before this date? It is not needed to put this data here. When you mention HIV for first time you should write the complete name of it as "Human Immunodeficiency virus".

Methods: why do you not include corticosteroids as immunomodulating drugs? Regarding underlying diseases you included most of them but not hepatic or other neurological diseases.

In line 168 you defined as negative controls those with negative antigen, why not those with negative PCR? could it lead a bias in selecting the controls?

When defining the ages, should I understand in line 177 that 0-1 year are children less than 12 months?

Results:

You say in line 186: "Most children (38.9%) did not have an immunocompromised..." Does it mean that most of the children included into the study had an immunocompromised state or underlying illness? If this is the case, you are selecting a specific population that is not representative of the general pediatric population, leading to a bias for the results.

In line 199 and 201 I would like to see 95%CI as shown before in lines 198.

Regarding the Wantai (+) children you selected 13/16 with symptoms, again this is not representative of pediatric population.

Discussion needs to improve before accept the manuscript.

Yours sincerely,

Staff Comments:

Preparing Revision Guidelines

To submit your modified manuscript, log onto the eJP submission site at <https://spectrum.msubmit.net/cgi-bin/main.plex>. Go to Author Tasks and click the appropriate

manuscript title to begin the revision process. The information that you entered when you first submitted the paper will be displayed. Please update the information as necessary. Here are a few examples of required updates that authors must address:

For complete guidelines on revision requirements, please see the Instructions to Authors at [link to page]. **Submissions of a paper that does not conform to Microbiology Spectrum guidelines will delay acceptance of your manuscript.**

Please return the manuscript within 60 days; if you cannot complete the modification within this time period, please contact me. If you do not wish to modify the manuscript and prefer to submit it to another journal, please notify me of your decision immediately so that the manuscript may be formally withdrawn from consideration by Microbiology Spectrum.

If you would like to submit an image for consideration as the Featured Image for an issue, please contact Spectrum staff.

Response to Reviewers

Reviewer #1

1. Thank you for this paper it was very interesting. When you compare your seroprevalence rates with the national seroprevalence study it may help to expand on the fact that their study was community based rather than your hospital based study. Your sample is very different as you have recruited from a hospital population rather than a representative paediatric population. The fact that you are seeing similar results both in the community and the hospital based seroprevalence studies is interesting. I think you make a good case for saliva antigens to be used in future research especially since this is more likely to be acceptable to parents of younger children who may be put off by a blood test.

Response:

We thank the reviewer for their comments and their acknowledgement of the potential of saliva antibody assays for children. We agree with the reviewer that we should emphasize the similarity in prevalence of our hospital population when compared to community based studies. In the Discussion section, we elaborated on the similar antibody prevalence despite the difference in study population. (page 12 lines 255 – 258)

Reviewer #2

1. Abstract: it is difficult to read, the information in the results need to be improved, i.e. lines 51 and 52 cannot begin with numbers of percentages.

Response:

We appreciate the reviewer's interest in our manuscript and thank the reviewer for their comments. We have rewritten the abstract results to improve the readability of this section. (page 2 lines 51 – 57)

2. Introduction: why do you comment the incidence of COVID-19 in December if the study was carried out before this date? It is not needed to put this data here.

Response:

We agree with the reviewer that the national incidence in December 2020 is not of additional value in this section. We aimed to provide the readers some background information on the national situation during recruitment of our study. We thus rephrased this sentence to better explain the context during the recruitment period. (page 4 lines 78 – 79)

3. When you mention HIV for first time you should write the complete name of it as "Human Immunodeficiency virus".

Response:

This sentence was rewritten according to the suggestion of the reviewer. (page 4 line 87)

4. Methods: why do you not include corticosteroids as immunomodulating drugs? Regarding underlying diseases you included most of them but not hepatic or other neurological diseases.

Response:

Corticosteroids were considered as immunomodulating drugs, although not reported in this section of the manuscript. We have added this information to improve the Methods section. (page 6 line 123)
Children with hepatic or neurological diseases were rarely included and were always primarily categorized as endocrine/metabolic, malignancy or psychomotor retardation.

5. In line 168 you defined as negative controls those with negative antigen, why not those with negative PCR? could it lead a bias in selecting the controls?

Response:

We acknowledge the sentence in line 168 was poorly phrased. We meant to indicate the presence of an assay control in the form of beads with no antigen to confirm for each individual that there is no nonspecific binding to the beads or blocking components. Additionally, we used healthy donor serum and saliva samples with antigen-coated beads to determine the assay background. We have added clarification to improve the methods section. (page 8 line 170) also to indicate the selection of the healthy donor samples.

6. When defining the ages, should I understand in line 177 that 0-1 year are children less than 12 months?

Response:

The youngest age group is indeed defined as children aged less than 12 months. We have rewritten the sentence to improve readability. (page 8 line 182)

7. Results: You say in line 186: "Most children (38.9%) did not have an immunocompromised..." Does it mean that most of the children included into the study had an immunocompromised state or underlying illness? If this is the case, you are selecting a specific population that is not representative of the general pediatric population, leading to a bias for the results.

Response:

We thank the reviewer for pointing out this sentence, as comorbidity was not clearly described. This sentence is rewritten to improve phrasing. (page 9 lines 190 – 192)
In addition, we elaborated upon the comorbidities in our population in the Discussion section, including the fact that our population is different from a community-based study. (page 12 lines 255 – 258)

8. In line 199 and 201 I would like to see 95% CI as shown before in lines 198.

Response:

Following the reviewer's suggestion we have added confidence intervals for lines 204 – 206.

9. Regarding the Wantai (+) children you selected 13/16 with symptoms, again this is not representative of pediatric population.

Response:

This sentence aimed to explain that in this population most children with antibodies in the Wantai either reported (mild) symptoms or close COVID-19 contacts. Our study was not powered to evaluate associations between antibody positivity and symptomatology as this was not our objective. Thus we did not intend to provide a representation of the pediatric COVID-19 population, nor conclude on any associations. We have rewritten this sentence to better explain that we describe characteristics. (page 9 line 211 – 212)

10. Discussion needs to improve before accept the manuscript.

Response:

We thank the reviewer for pointing this out and have made improvements to the discussion in several sections.

August 12, 2021

Dr. Maya W. Keuning
Amsterdam UMC
Amsterdam
Netherlands

Re: Spectrum00731-21R1 (Saliva SARS-CoV-2 antibody prevalence in children)

Dear Dr. Maya W. Keuning:

I am please to report that your manuscript has been accepted and I am forwarding it to the ASM Journals Department for publication. You will be notified when your proofs are ready to be viewed.

Sincerely,

Jennifer Dien Bard
Editor, Microbiology Spectrum
